# Objective monitoring of loneliness levels using smart devices: A multi-device approach for mental health applications

**Salar Jafarlou**[1]*, **Iman Azimi**[1,2], **Jocelyn Lai**[3], **Yuning Wang**[4], **Sina Labbaf**[1], **Brenda Nguyen**[3], **Hana Qureshi**[3], **Christopher Marcotullio**[3], **Jessica L. Borelli**[5], **Nikil D. Dutt**[1], **Amir M. Rahmani**[1,2,6]

**1** Donald Bren School of Information and Computer Sciences, University of California, Irvine, Irvine, California, United States of America, **2** Institute for Future Health, University of California, Irvine, Irvine, California, United States of America, **3** Department of Psychological Science, University of California, Irvine, Irvine, California, United States of America, **4** Department of Computing, University of Turku, Turku, Finland, **5** Department of Cognitive Science, University of California, Irvine, Irvine, California, United States of America, **6** School of Nursing, University of California, Irvine, Irvine, California, United States of America

* jafarlos@uci.edu

**Data Availability Statement:** The data used in this study include sensitive health information, and the informed consent signed by the participants does not allow the data to be made publicly available due

## Abstract

Loneliness is linked to wide ranging physical and mental health problems, including increased rates of mortality. Understanding how loneliness manifests is important for targeted public health treatment and intervention. With advances in mobile sending and wearable technologies, it is possible to collect data on human phenomena in a continuous and uninterrupted way. In doing so, such approaches can be used to monitor physiological and behavioral aspects relevant to an individual's loneliness. In this study, we proposed a method for continuous detection of loneliness using fully objective data from smart devices and passive mobile sensing. We also investigated whether physiological and behavioral features differed in their importance in predicting loneliness across individuals. Finally, we examined how informative data from each device is for loneliness detection tasks. We assessed subjective feelings of loneliness while monitoring behavioral and physiological patterns in 30 college students over a 2-month period. We used smartphones to monitor behavioral patterns (e.g., location changes, type of notifications, in-coming and out-going calls/text messages) and smart watches and rings to monitor physiology and sleep patterns (e.g., heart-rate, heart-rate variability, sleep duration). Participants reported their loneliness feeling multiple times a day through a questionnaire app on their phone. Using the data collected from their devices, we trained a random forest machine learning based model to detect loneliness levels. We found support for loneliness prediction using a multi-device and fully-objective approach. Furthermore, behavioral data collected by smartphones generally were the most important features across all participants. The study provides promising results for using objective data to monitor mental health indicators, which could provide a continuous and uninterrupted source of information in mental healthcare applications.

to ethical restrictions. According to the current approval by the Institutional Review Board of the Human Research Protections program at the University of California, Irvine, participants gave permission to use the collected data only for the purposes described in the consent. Data requests may be subject to individual consent and/or ethics committee approval. Researchers wishing to use the data should contact the Human Research Protections program at the University of California, Irvine. Contact details are as follows: (949) 824-8170, email: IRB@uci.edu. We recommend contacting the principal investigator (PI) of the research project first. Associate Professor Amir Rahmani can be reached at 2535 Nursing and Health Sciences Hall, Irvine, CA, email: a.rahmani@uci.edu.

**Funding:** The author(s) received no specific funding for this work.

**Competing interests:** Enter: The authors have declared that no competing interests exist.

## Introduction

### Loneliness and social isolation—A public health epidemic

Though frequently used and described in the same contexts, loneliness is distinct from social isolation. Loneliness refers to a subjective feeling wherein the individual experiences a lack of actual connection within social interactions [1, 2]. It is also important to note distinctions between loneliness from being alone, in that individuals may be alone but not feel lonely. Loneliness has been defined as a universal human experience, yet highly subjective in the way it is felt [3, 4]. Social isolation, on the other hand, refers to the specific lack of social interaction or access to one's social networks [5]. A further distinction to consider is that loneliness is separate from an emotion, and can be viewed as a judgment about one's social connections or lack thereof, that elicit negative emotions and feelings [3, 4]. Being alone or experiencing social isolation may not include the subjective adverse negative feelings or judgment of one's social relationships that loneliness entails.

A substantial number of researchers have delved into diverse approaches to accurately define loneliness as an accessible issue. Sullivan proposed that individuals may not be aware of their own or others' states of loneliness or social deficiency because of a number of factors, including defense mechanism, lack of self awareness and cultural norms [6]. Young highlighted the temporal aspect of loneliness, suggesting that studies that dynamically collect individual information over extended periods offer exceptional reference value [7]. He argued that loneliness is a dynamic experience that can fluctuate over time, and that in order to fully understand it, we need to study it longitudinally. Rook characterized loneliness as an ongoing emotional turmoil experienced in scenarios of isolation, miscommunication, or rejection, marked by an insufficient level of meaningful social engagement and participation in uplifting activities. Alleviation can be achieved through avenues that foster emotional intimacy. He defined *loneliness index* based on six items assessing different aspects of loneliness [8].

Both loneliness and social isolation have profound health implications and are linked with increased morbidity and mortality rates [9, 10]. Concerns for the health implications of loneliness and social isolation were heightened during the COVID-19 pandemic [11, 12], during which people around the world were requested to minimize contact and comply with stay-at-home orders to reduce transmission of the disease.

When focusing on loneliness, individuals with lower socioeconomic status, younger adults, and students in particular were found to have high feelings of loneliness during the pandemic [13, 14], though empirical evidence regarding the differences in loneliness over time and the psychological implications of lockdown remain unclear [15–17]. However, a recent meta-analysis by Ernst and colleagues [18] suggested a small yet significant effect size in the differences between feelings of loneliness before and after the pandemic. Several researchers argue that there are differential associations between loneliness and social isolation with health outcomes, such that loneliness contributes more to mental health outcomes while social isolation is more important in predicting declines in cognitive and physical health [19–22]. However, other studies suggest that the differential health implications of loneliness and social isolation are much more difficult to disentangle; for example both constructs are associated with increased risk for cardiovascular disease and interact in their relation with mortality [5, 23, 24]. In summary, given their crucial link to health-risks, both loneliness and social isolation are necessary to address. To target ways of reducing these feelings, it is important to first determine when and how they occur.

## Assessments and predictions of loneliness and social isolation in daily life

Although a robust body of literature has established a strong link between loneliness and health outcomes [25, 26], this work is constrained by the fact that the majority of this work has relied on self-reported measures of loneliness and social isolation that require that participants report on their symptoms spanning an extended period of time (e.g., in the past two weeks). It can be difficult to accurately report on symptoms over such a long time period, and such symptom reports can be influenced by current mood [27–29]. An on-going and exciting avenue to examine loneliness, both its occurrence and association with health and social outcomes, is through its occurrence in daily life [30]. By assessing loneliness with greater ecological validity, we may be able to understand how and when loneliness plays a role in physical and mental health outcomes. For example, existing literature has found that loneliness in daily life is linked to increased feelings of negative affect, greater negative appraisals of social interactions, and more time spent alone; furthermore, these associations had lagged associations with loneliness and depression at a later time point [30, 31].

In addition to outcomes associated with loneliness, several studies have been conducted to assess the feasibility of detecting and modeling loneliness and social isolation in real-world settings. Previous approaches using intensive longitudinal self-reports can put a heavier load on users, resulting in unsatisfactory experiences for the users due to the reliance on frequent engagement [32, 33]. An attempt to resolve this issue involves leveraging users' behavioral data as collected by continuous mobile sensing. Using continuous and objective sensing devices (e.g., smartwatches) to monitor loneliness allows for a more accurate and representative assessment when taking into account contextual factors in a user's environment. Smartphones can monitor certain aspects of a user's life using multiple embedded sensors (e.g. GPS, accelerometer, microphone, light sensor and phone usage logs). Furthermore, passive sensing can quantify the behavioral patterns of participants, including the type and frequency of activities performed, changes in geographic location, and phone usage. Such work has already begun to leverage this type of data to determine features that contribute to loneliness with high predictive accuracy [34–37]. For example, mobility and digital media use are found to be features linked to feelings of loneliness [32, 38]. Additionally, the presence of other people and differences in phone usage were also shown to be predictors of loneliness [38, 39]. Doryab and colleagues [38] have attempted to classify loneliness levels amongst college students before and after a semester by leveraging data collected with smart wristbands and phones. Using these devices, Doryab and colleagues were able to extract features about calls, locations and screen usage collected from phones along with sleep and activity level features from smart wristbands. Nevertheless, the loneliness scale labels obtained in this study were limited to only two time-points: pre- and post-semester, which obscures the temporal dynamics of the loneliness data. The wristband used in this study (Fitbit Flex 2) assessed sleep information using the same motion sensors used for activity level recognition. This is while recent devices are capable of leveraging heart rate and other physiological sensors for sleep quality assessment [40, 41].

Despite existing literature that has leveraged the capabilities of smartphones in monitoring loneliness and well-being more broadly, a few novel directions in this approach have yet to be fully explored. First, ubiquitous sensing and advances in technology can allow us to potentially monitor and predict loneliness with higher granularity. Most studies have assessed loneliness as a trait measure or with few assessments throughout the study duration [32, 33, 38, 39] as opposed to intensive assessments of loneliness across multiple days over a long period of time. For example, work by [39, 42] measured loneliness, social anxiety, and affect at the beginning and end of their study. Although measuring these levels at two time points can provide general trends, such approaches likely omit the temporal changes in loneliness during the course of

the study or at the daily level. There are, however, a few studies that have assessed loneliness and mental health states multiple times a day [43, 44]. In a study by Ben-Zeev et. al. [43], participants completed a 10-week study that passively collected their speech patterns, location, kinesthetic activities, and sleep duration using smartphones alongside self-reported assessments of daily stress, depression, and loneliness. StudentLife [44], was another study conducted by Wang and colleagues [44], participating 48 students during a 10 week period collecting various contextual data from phone and comprehensive ecological momentary assessments (EMA) assessing variables including stress, mood, and loneliness etc. The authors performed detailed analysis on the correlation of contextual information extracted by smartphones and different EMA data; however, they did not report loneliness detection/classification tasks. While these works provide a relational analysis between the loneliness scales and data, our study aims to take a predictive approach to loneliness.

Lastly, an exciting avenue for the field is the use of a comprehensive approach to monitor and predict loneliness and mental states that involves multiple modalities of passive sensing. A limitation of previous work that relies on passive sensing using smartphones is the expectation of continuous user engagement. While individuals do frequently access their smartphones, there may be times where data collection through smartphones may lack granularity in assessment (e.g., leaving smartphones in bags while at the gym or doing an activity). With the progression of technological tools for entertainment, users, particularly younger adults, may use other modalities for their online presence, such as tablets, game consoles, and smartwatches. Furthermore, smartphone assessments are often limited to assessment of behaviors, unlike other ubiquitous devices such as smart wearables that can provide physiological state measurements. Indeed, prior literature that has used smartphones in real time typically focuses on usage of phone features (e.g., incoming and outgoing calls or messages), geolocation, and participant self-reported responses to assess how these behaviors relate to health and social outcomes [42, 44]. Incorporating other ubiquitous devices such as the OuraRing and Samsung smartwatch allows for the continuous collection of several features such as sleep (e.g., sleep duration, quality, restlessness) and cardiovascular (e.g., heart-rate, heart-rate variability, blood pressure) indicators [40]. The ability to assess sleep and cardiovascular indicators is important, given existing literature suggesting the links between loneliness with sleep disturbances and lower cardiac output [31, 45, 46]. Moreover, the effectiveness of each of these devices in modeling loneliness levels in individuals is a research gap that has been inadequately explored. Assessing the strengths and limitations of different passive sensing tools in predicting loneliness not only is important for determining best practices and feedback for tools to improve their predictive accuracy, but could also enable us to gain a more profound understanding of how various behavioral and physiological features impact the user's loneliness and in the future, mental health more broadly. It would also assist in optimizing the modeling and inference processes across various dimensions, such as accuracy and energy consumption, among others. Therefore, leveraging passive sensing using multiple devices with the capacity to detect unique factors such as sleep and cardiovascular activity would build upon existing literature to provide a more comprehensive and granular understanding of when and how loneliness occurs.

## Current research

Altogether, the intersection of mental health and technology offers an exciting opportunity to build upon foundational work in the field [38, 44, 47] by developing a more comprehensive monitoring framework capable of assessing the occurrence and predictability of loneliness. In this study, we develop a method of passively and continuously monitoring loneliness using

smart wearables and smartphone devices. Using these devices together allows us to assess a wide range of behavioral patterns such as phone usage, communications, and locations as well as physiological patterns such as sleep indices and cardiovascular data. We leverage these devices to propose and evaluate a fully objective loneliness detection method that does not rely on user engagement. We also aim to examine which behavioral and physiological features were most predictive of loneliness for each individual. Finally, our third aim is to determine device efficiency of loneliness monitoring and detection.

## Method

### Participants

Full-time college students (N = 30) at a west coast university between the ages of 18–22 were recruited to participate in an intensive longitudinal study investigating loneliness and mental health. Participants were eligible if they spoke fluent English and used an Android smartphone compatible with the Oura Ring and Samsung Active 2 watch. Students were ineligible to participate if they were married, had children, were returning to school after a three year hiatus, or if they experienced any severe forms of psychopathology (i.e., diagnosed with clinical depression or substance use disorders, psychosis, or any form of suicidal ideation). The exclusion criteria were intended to ensure a relatively homogenous sample of college students and individuals in emerging adulthood. We reasoned that students whose demographics were different (parents, older adults, and individuals returning to school) may have a different experience than emerging adults as they develop their identity and social networks. Students were recruited during September 2021 until January 2022 through faculty outreach where professors may then make announcements to their classes. Social media posts on institutional platforms (e.g., Facebook, Reddit, and Discord) were also used to recruit students. Interested participants reached out via email and were administered a screening survey to assess for moderate to severe depression or suicidal ideation. Individuals who met these criteria were then reached out by the clinical psychologist on our team (JB) for additional support and follow-up. Campus and local wellness resources were provided to all individuals who reached out or completed the screening survey.

The study participants comprised a diverse group in terms of gender, with 57% identifying as female, 40% as male, and 3% as non-binary. In relation to race, the sample exhibited a balanced distribution, with 40% identifying as Asian, another 40% as Hispanic/Latino, 13% as White, and 7% as biracial. Participants also spanned multiple school years, with 17% being in their first or second year, 40% in their third year, 23% in their fourth year, and 3% in their fifth year. The mean age of the participants was 19.90 years, with a standard deviation of 1.21, reflecting the age diversity within the cohort. The data collection occurred simultaneously among participants, spanning from January to March 2021.

### Ethics approval

This research received approval from the Institutional Review Board (HS#2019–5153) overseen by Dr. Elizabeth Cauffman, at the University of California, Irvine. Prior to their involvement, participants gave written consent. As part of a broader investigation into mental health trajectories, potential participants underwent screening for severe depression or suicidal thoughts. A clinical psychologist on the project team provided consultation during this screening. Participants had the option to withdraw from the study, and the principal investigators had the authority to halt their participation at any time to safeguard their well-being.

## Procedure

For the purpose of this investigation, only relevant procedures and our monitoring phase will be described from the larger study. If participants met the inclusion criteria, they were then contacted to schedule their first in-person lab session in which they completed a baseline battery assessment of mental health, emotions, well-being related measures and a written consent. The mental health assessment included measures of psychopathology, on the basis of which participants were withdrawn from the study if the research team determined in consultation with JB that continued participation would be unsafe. During the first lab session, Oura Ring, Samsung Gear Sport watch, and smartphone apps related to the study were set up and downloaded for the participants. Participants were then instructed on how to use the devices and were guided through tasks that they would be completing for the next four weeks of their participation. Participants were instructed to wear their devices at all times throughout the study period, except under certain conditions (e.g., while charging devices or while performing intensive activities that risked damage to the watch). For the smartphone apps, participants downloaded AWARE that monitored their phone usage and a study-designed app, mSavorUs, which would prompt individuals to complete a brief survey five times per day, using an interval-based ecological momentary assessment (EMA) design. The survey included questions about affect, interactions with others, and feelings of loneliness, social isolation, and connectedness. Apps relating to the wearable devices were also downloaded onto participants' phones for proper device functionality. Participants completed this monitoring phase for about 8-weeks and were compensated for their involvement in all components of the study.

## Data collection

To support our longitudinal study design, we built a platform capable of collecting and storing data efficiently [48]. We designed a research dashboard that allowed the researchers to have constant access to the data being collected in order to monitor participants' progress. The dashboard visually displayed the collected data and a summary of each participant's daily activities. The information was then used to track and assess possible connectivity issues with the devices throughout the study duration. The collected data fell under three categories: subjective self-report questionnaires (used as target labels), objective physiological data, and objective behavioral data. Participants' objective physiological data was collected using the Oura ring and Samsung watch. The Oura ring was used to assess information regarding the participant's sleep patterns, and the Samsung watch was employed to conduct continuous physiological assessments throughout the day. The objective behavioral data was collected using the AWARE phone application. AWARE provides us with mobile sensory data and participants' phone usage details, which can then be used to derive behavioral features. Subjective self-report data was collected using the mSavorUs phone application that was developed for the purpose of collecting subjective experiential data and delivering interventions in this study.

**Subjective assessments of loneliness using the mSavorUs phone application.** Participants completed subjective assessments of loneliness, emotions, and social interactions using the mSavorUs app. This was to reflect our EMA design where participants were prompted five times per day, with additional perceived sleep questions in the morning EMA. The five assessments were scheduled to be sent within a 4-hour time window of each other. Participants were notified on their phones to complete the brief survey. Questions included in the EMA assessed how positive and negative participants felt in the moment. For loneliness, participants were asked, "How lonely do you feel right now?". Other questions included asked about how connected and socially isolated they felt, and if they recently interacted with people, how many

and with whom. The inputs from participants on loneliness were assessed on a scale of 0–100, which were subsequently converted into labels for the purpose of classification.

**Objective physiological data using samsung watch and oura ring.** The Oura Ring offers a range of metrics regarding the user's physical activity, and sleep patterns [12]. However, Photoplethysmography (PPG)-based wearable devices, including the Oura Ring, can encounter noise during signal collection, particularly when utilized in a home-based monitoring setting. For our purposes, we used data from Oura that included the user's sleep, physiology and physical activity. The final list of selected data included heart-rate related information during sleep, the estimated time of start of sleep after getting into bed, as well as the number of minutes with high, medium and low level of physical activity during the day (refer to S3 Table in S1 Appendix for a detailed list of features of Oura ring used in this study). Samsung Active 2 watch operated on the Tizen open-source operating system [49], allowing for the development of custom data collection applications. This open-source platform enabled us to develop an application specifically designed for the watch to collect the data in a manner customized to the purposes of our study. The watch app activated every two hours to collect twelve minutes of PPG signals, used to extract heart rate and heart rate variability. The collected data was then synchronized with our server through wifi or bluetooth.

**Objective behavioral data using the AWARE app and subjective experiential data using mSavorUs.** For the objective behavioral data, we used the AWARE app available for Android smartphones. This app's sensors were configured to collect participants screen usage (e.g., lock, unlock time), application usage (e.g., notification, time of opening any app), their keyboard strokes pattern, battery level, communications (e.g., message/calls sending and receiving time) and GPS location [50]. Finally, for subjective experience, we designed and developed the mSavorUs application as a portion of a larger intervention-based study. This application [51] included assessments of loneliness and social isolation using mobile app interfaces and push notifications.

## Data analytic plan

**Feature extraction.** We developed Heart Rate (HR) and Heart Rate Variability (HRV) extraction methods from the raw PPG signals collected by the Samsung watch. The raw PPG signals are highly susceptible to environmental noise motion artifacts. To illustrate the problem, we plot two 60-second PPG signals. Fig 1(a) is a clean PPG signal, showing heart beat oscillations. However, Fig 1(b) is a noisy PPG, distorted by the subject's hand movements. Such distorted signals will result in unreliable HR and HRV features extraction. Therefore, we developed a PPG processing pipeline to address this problem.

The pipeline includes 3 main stages: signal quality assessment (SQA), signal reconstruction, and PPG peak detection. The SQA classifies PPG signals as "clean" or "noisy" by extracting five features from the signal, including interquartile range, standard deviation of the power spectral density, range of energy of heart cycles, average Euclidean distances, and average correlation between a template and heart cycles [52]. After we performed SQA, short-term "noisy" segments (less than 15 seconds) were reconstructed using a trained generative adversarial network (GAN) model [53]. The GAN model was trained to reconstruct noisy PPG using the information both in the distorted part and its proceeding clean signals. Then, a trained Dilated Convolution Neural Network (DCNN) was employed to detect the systolic peaks [54] and inter-beat intervals (IBI). Finally, HR and HRV-related features were extracted from the IBI signals. We access the Oura ring data through an application programming interface (API) that provides the processed features of sleep and activity. No further feature

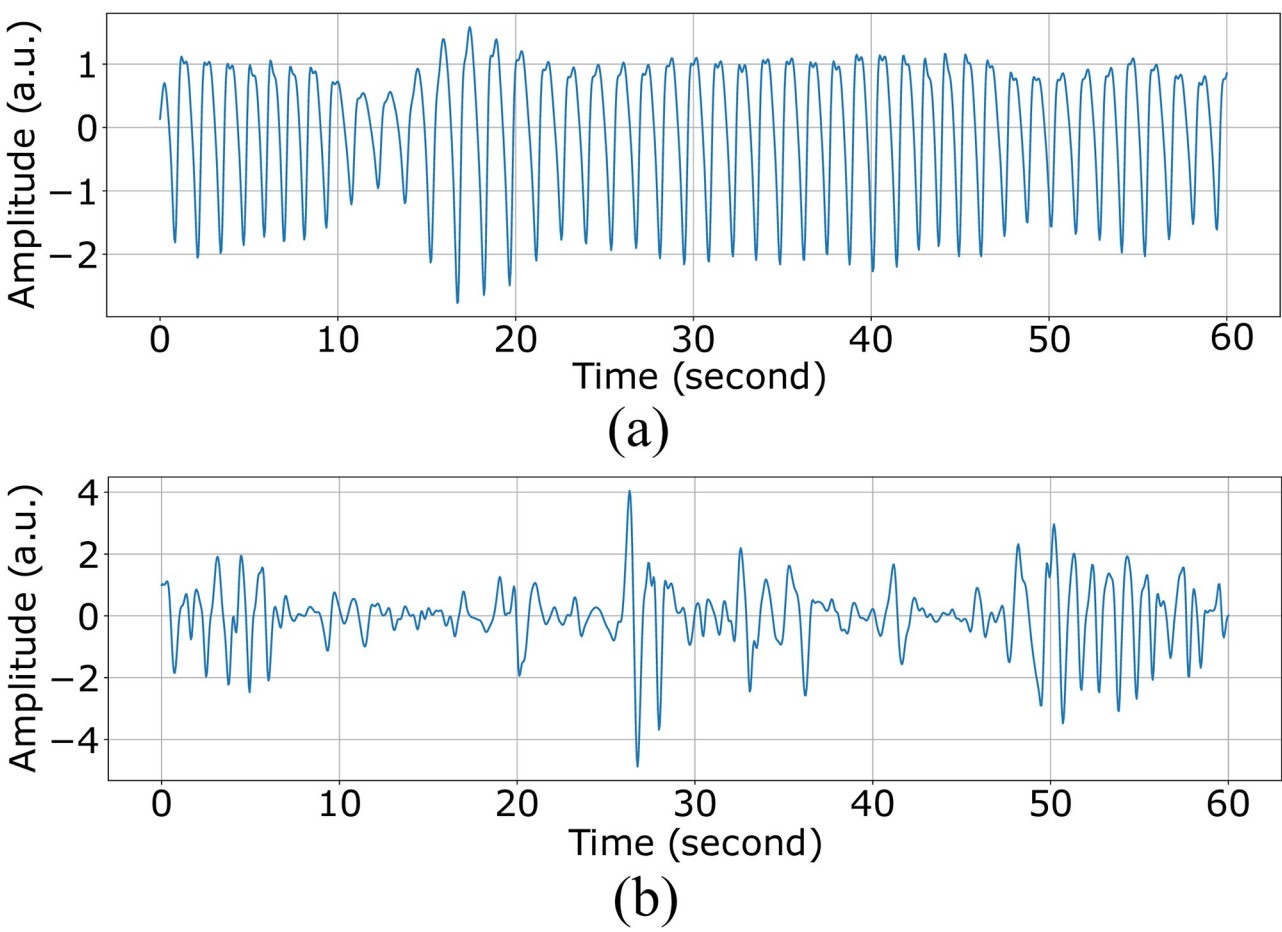

**Fig 1. 60-second PPG segments (a) clean (b) distorted by hand movements.**

extraction or post-processing steps were done on the ring data (please refer to S1 and S2 Tables in S1 Appendix for the full list and description of the features extracted from ring and watch).

We developed methods to extract behavioral parameters of the participants' daily living through the longitudinal behavioral data collected by the smartphones. For calls, we used the duration sums and number of calls in each category (e.g., outgoing, incoming and voicemail). For location data we first located the participant's house as the most frequently visited place during the nights of study. Then given a location time window, we extracted variance of latitude, variance of speed, mean of speed, number of places, home duration, outdoor duration, mean of outdoor duration ($> = 2$ places), standard deviation of outdoor duration ($> = 2$ places), longest duration type other than home ($> = 2$ places), and total travel distance($> = 2$ places). For messages and notifications, we counted the number of messages in each category. Table 1 demonstrates all of the categories and three samples for each of them (please refer to the S3 Table in S1 Appendix for a detailed list of all features extracted from the phone).

Some of the features from the devices were collected in a single assessment per day (i.e., sleep related features) whereas others were assessed with higher resolution, multiple times throughout the day (i.e., location, phone lock-screen, and PPG recordings; for a list of all features, see S1–S3 Tables in S1 Appendix). For data that was sampled multiple times per day, we tested different time windows (e.g., from 4 to 48 hours) for aggregating the values in

**Table 1. Different categories of app notifications and with three examples of each.**

| Apps Category | App Examples | | |
|---|---|---|---|
| Productivity | Google Doc | Office Outlook | Gmail |
| Photography | Android Photos | Bazaar | Lightroom |
| Communication | Whatsapp | Android Messenger | Discord |
| Lifestyle | Oneconnect | T-Mobile Tuesdays | Samsung pay |
| Auto & Vehicles | Gearhead | Toyota | Carfax |
| Travel & Local | Couch Surfing | Google Maps | Uber |
| Education | duolingo | Wonder | Canvas |
| Finance | mint | American Express | SplitwiseMobile |
| Video Players & Editors | Vlc player | Youtube | Capcut |
| Social | Snapchat | Katana | TikTok |
| Books & Reference | audible | Scribd | Chirp |
| Shopping | Slickdeals | Amazon | Target |
| Health & Fitness | Fitindex | Myfitnesspal | shealth |
| Entertainment | Amazon Prime Video | Netflix | Hulu |
| Business | Duo | Slack | LinkedIn |
| Music & Audio | Spotify | Youtube Music | Soundcloud |
| Tools | Sprint | Android search box | Google Mobile Service |
| Unknown | Gametools | Clock | Calendar |

order to pair with the subjective responses (i.e., self-reported loneliness used for labeling). Fig 2 shows how different window lengths for each modality have been used to compile data records. After extracting mentioned feature values given a time window, we selected the optimum time window for each feature based on the correlation with the target questionnaire response values.

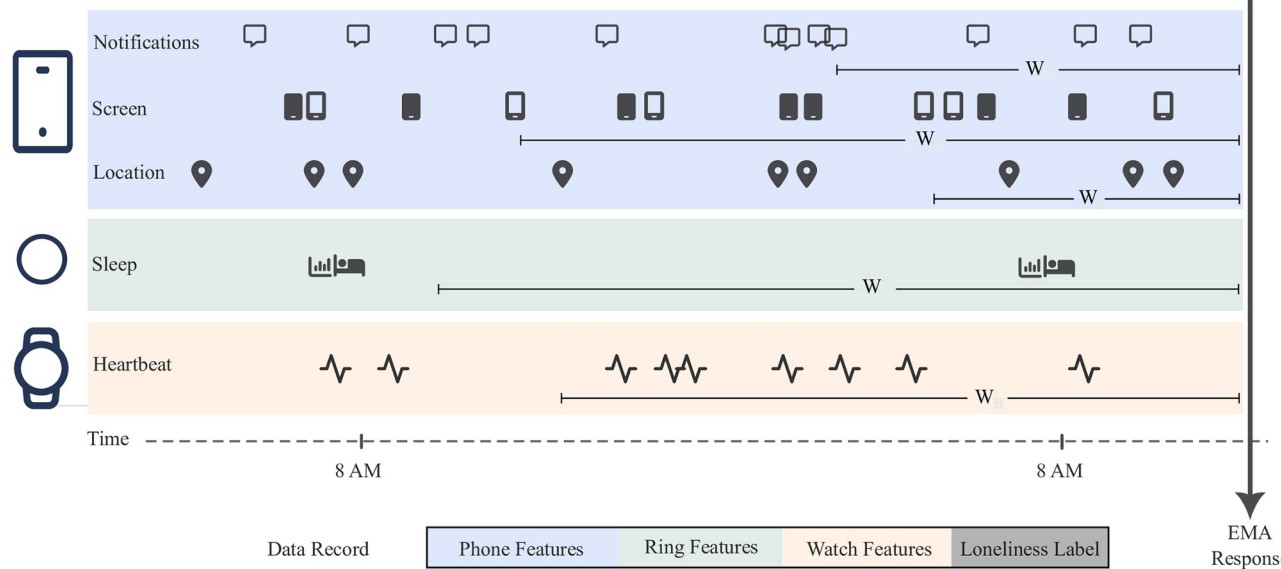

**Fig 2. Schematic presentation of feature extraction window and data record compilation from modalities.** Each modality provides different sensors and separate window lengths could be defined for extracting features from each of these sensors.

## Missing data

For intensive longitudinal studies assessing experiences in real-time, many factors could interrupt continuous data collection and result in missing data. These factors may result from participants forgetting to wear their devices or charge them overnight, or may result due to technical issues (e.g., server congestion or permission issues over the phone). Using the monitoring tools in the data collection server, we were able to track most of these issues over the course of study and solve them in the shortest time possible. After all these arrangements, having a certain degree of missing data is inevitable and must be taken care of in the analysis stage. We removed the features that had more than 30% missing data. Because of varying windowing policies for different features, it was necessary to apply the dropping criteria to each feature individually, rather than to each modality as a whole. In total, 19 features were omitted due to a significant amount of missing values. In our analysis, we tested two data imputation methods to handle missing data. We replaced the missing values with the average of: A) two preceding and succeeding valid samples, or B) all valid values [55] of that participant. The latter method yielded a higher correlation with the target questionnaire response values for the majority of the features. Therefore, for consistency, we used this method to handle missing data for all of the features.

## Classification

We defined the binary classification labels (true/false values) based on the participant's self-reported loneliness rating (scaled from 0–100, 0 = not at all, 100 = extremely) as below or above the median to have a balanced number of labels. We developed a random forest [56] method to predict these classification labels using the 84 selected features. Random Forests is a machine learning algorithm that combines multiple decision trees to make more accurate predictions. Leveraging multiple decision trees enables this algorithm to process high dimensional data efficiently. Before training predictive models, we Z-normalized the features using each participant's data in the training set to reduce interpersonal bias in feature space. For model evaluation, we used a one-subject-out policy for predictive modeling evaluation, where we used the most recent 50% of each participant's data as a test set and the remaining data of that participant with all the other participants for the training model. We calculated the accuracy, F1-score, precision, recall, and mean squared error for all the test entries combined. We defined true positive and false positive predictions if our model detected the loneliness class respectively correct and incorrect (according to EMA labels). Similarly, true negative and false negative predictions were instances where the model did not detect the loneliness class while they were correct and incorrect, respectively. Given these terms we can define aforementioned performance metrics as:

$$Accuracy = \frac{number\ of\ true\ negative + number\ of\ true\ positive}{total\ number\ of\ data\ records}$$

$$Precision = \frac{number\ of\ true\ positive}{number\ of\ true\ positive\ +\ number\ of\ false\ positive}$$

$$Recall = \frac{number\ of\ true\ positive}{number\ of\ true\ positive\ +\ number\ of\ false\ negative}$$

$$F1 = \frac{2 * precision * recall}{precision\ +\ recall}$$

$$Mean\ Squared\ Error = \frac{1}{N}\sum\left(num\ of\ false\ negative\ +\ num\ of\ false\ positives\right)^2$$

These reporting metrics are important for data interpretation because they can help us to understand the strengths and weaknesses of a classification model. For example, a model with high accuracy may have low precision or recall, meaning that it is good at predicting the majority class but not good at predicting the minority class. Or higher precision and lower recall means that the model is more likely to predict positive instances correctly, but it is less likely to identify all positive examples. In our case this means the model detects the majority of loneliness instances in participants but not all of them.

### Feature importance

A limitation of using classic model evaluation metrics is that they lack insight into the dynamics of prediction. To address this limitation, the SHapley Additive exPlanations (SHAP) method was used to investigate the contribution of features to predictions given a model [57]. SHAP offers explainability and insight into the contribution of each feature used by a given model in the prediction stage. Given a trained model, this method generates numerical estimates (called SHAP values) that represent how much a feature value affects the output value of the model and what direction (toward either of the classification labels) this effect is. Variations of this method have been proposed for different machine learning models. Some of them consider the trained model as black-box and extract local explanations using efficient sampling in the feature space [58]. Other variations of SHAP analysis methods are proposed for specific machine learning models. In this work, we leveraged path dependent feature perturbation algorithms [59] developed for tree-based models. This method splits the feature space while recursively following the decision path for a given datapoint. This approach enabled us to enhance the accuracy and interpretability of our models, making them more effective tools for decision-making and analysis.

### Results

#### Aim 1: Loneliness detection overall performance

We first present the performance of our proposed loneliness detection method enabled by using the multiple ubiquitous sensing devices AWARE, Oura ring, and Samsung watch. As previously mentioned, we conducted a two-month monitoring study with 30 participants who completed five self-report assessments per day, totaling to about 7,300 data points. The trained model was evaluated by comparing the estimated loneliness values with the corresponding ground truth values (i.e., collected by subjective questionnaires). This procedure was repeated for every participant; in other words, 30 different personal models were built. In the following, we report the aggregated results obtained from all the trained models. Our loneliness detection models obtained an accuracy of 82% Table 2, precision score of 81%, and value of 0.83 Area Under Curve (AUC). Higher AUC generally represents better classification capability (ranging

**Table 2. The performance of the loneliness detection methods using different devices.**

| Device(s) | Accuracy | Precision | Recall | F1 score | MSE |
|---|---|---|---|---|---|
| Oura Ring | 0.565 | 0.556 | 0.988 | 0.711 | 0.434 |
| Samsung Watch | 0.781 | 0.809 | 0.780 | 0.794 | 0.220 |
| AWARE (Smartphone) | 0.810 | 0.877 | 0.756 | 0.812 | 0.189 |
| Samsung Watch + Oura Ring + Aware | 0.822 | 0.883 | 0.773 | 0.824 | 0.177 |

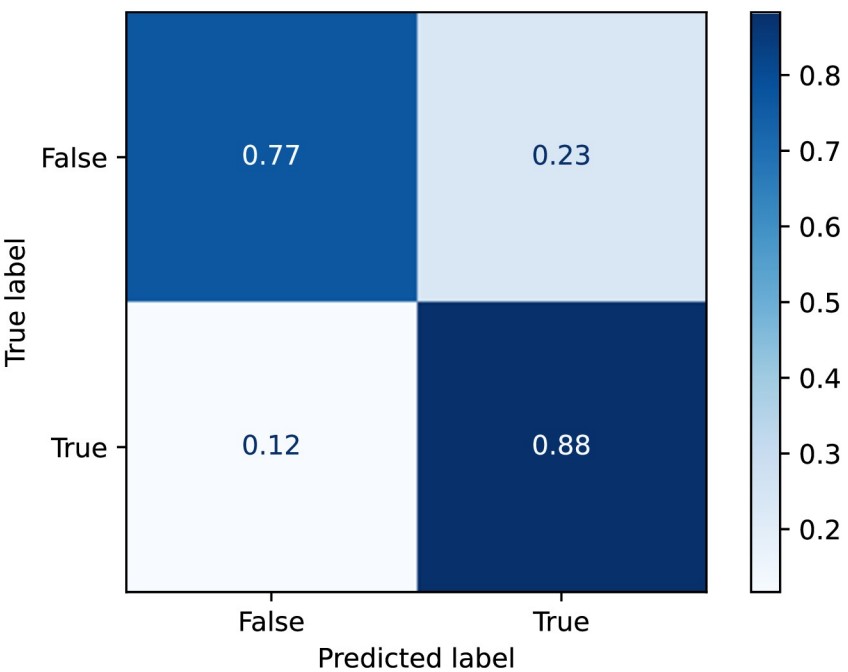

**Fig 3. Confusion matrix of loneliness prediction model performance using all channels.**

from 0.5 showing random classification to 1 that is for a perfect classification). Fig 3 shows the confusion matrix of the loneliness detection models extracted from about 3,600 tested samples. The obtained true positives (detecting loneliness correctly) and true negatives (estimating no loneliness correctly) were considerably lower compared to the false positives and false negatives.

### Aim 2: Explainability and feature importance analysis

As discussed, we obtained loneliness detection accuracy of 82% using all the modalities from three devices (smartwatch, ring, and phone). In this section, we explore further into the analysis to investigate the effect of the features on the performance of loneliness detection using explainable machine learning techniques. To this end, we carried out SHapley Additive exPlanations [57] (SHAP) analysis to gain deeper insight into the detection models. SHAP values serve as a measure showing the significance of each feature in the model's ability to perform the detection. As previously mentioned, we trained a personal model for each participant. To assess the impact of each feature on the loneliness detection among participants, we calculated the mean SHAP values across the test samples.

Fig 4 depicts the absolute SHAP values of the 20 features for each participant, with dark blue and light green hues showing highest and lowest values, respectively. Overall, the 6 most influential features from our prediction model were related to behavioral markers extracted from smartphones (i.e., the AWARE platform), while the HRV features—collected from the smart watch—had less impact. However, the influence of each feature varied across participants. For example, the number of notifications from lifestyle and communication applications had a significant impact on Participants 15 and 20, respectively, whereas these features did not exert as similar of an impact on the remaining participants. This inter-individual difference indicates the importance of personalization in loneliness detection methods.

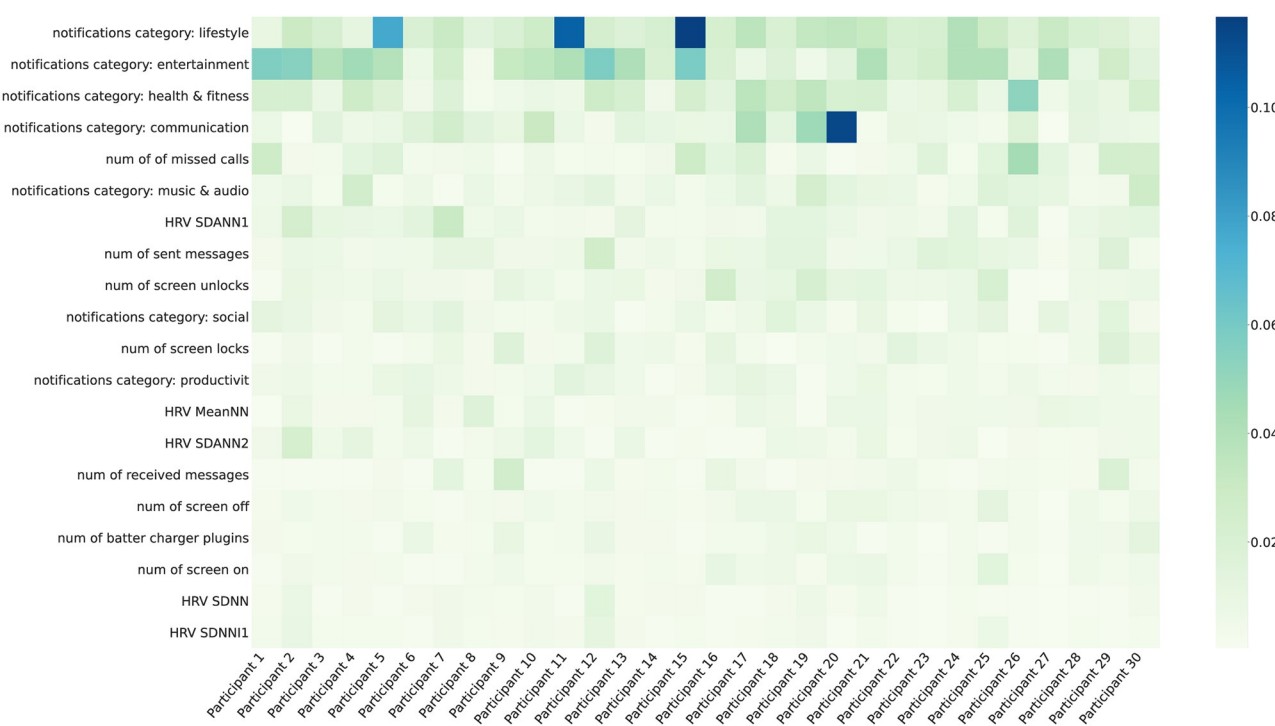

**Fig 4. Mean of absolute SHAP values for each participant-feature pairs.** The top 20 features with the highest average values across participants are presented in a sorted order. *HRVSDANN*1, *HRVSDANN*1 is the standard deviation of average RR intervals extracted from segments of PPG collected by the smart watch for 1-minute and 5-minute intervals respectively. Please refer to S1 Appendix for a full description of all features.

The findings shown in Fig 4 further offer a broad understating of the impact of individual features on the loneliness detection. However, due to averaging the absolute values, this representation obscures the distribution and direction of the effects of each feature on each participant. Specifically, a positive SHAP value denotes that the corresponding feature shows a positive impact on the loneliness class detection, while a negative value contributes to the non-loneliness class detection. Thus, Fig 5a and 5b shows the positive and negative SHAP values of two randomly selected participants, revealing the variation in the importance of the features. Loneliness class is represented on the right side of the y axis in this Figure, while non-loneliness is represented on the left side. The red and blue colors illustrate negative associations and positive associations respectively for a given feature. For example, physiological features were the most influential for Participant 8 (Fig 5a), whereas Participant 10's most important features (Fig 5b) were behavioral features. It also shows that on average, Participant 8's lower values of HRV metrics (represented by the blue color) are associated with higher feelings of loneliness (left side of the vertical line). These figures also indicate how the order and impact of features vary between the participants for loneliness detection. For example, for Participant 8, the most influential features are extracted from the smartwatch. However, this effect is less evident for Participant 10 (Fig 5b). For instance, decreases in the HRV CVSD feature (presented in blue color) results in loneliness for Participant 10. This is while higher values of this feature (red) results in non-loneliness class.

## Aim 3: Loneliness detection using different sets of devices

We investigated the effects of different modalities on the performance of our loneliness detection method. To this end, in the training phase, we built four random forest models, three of

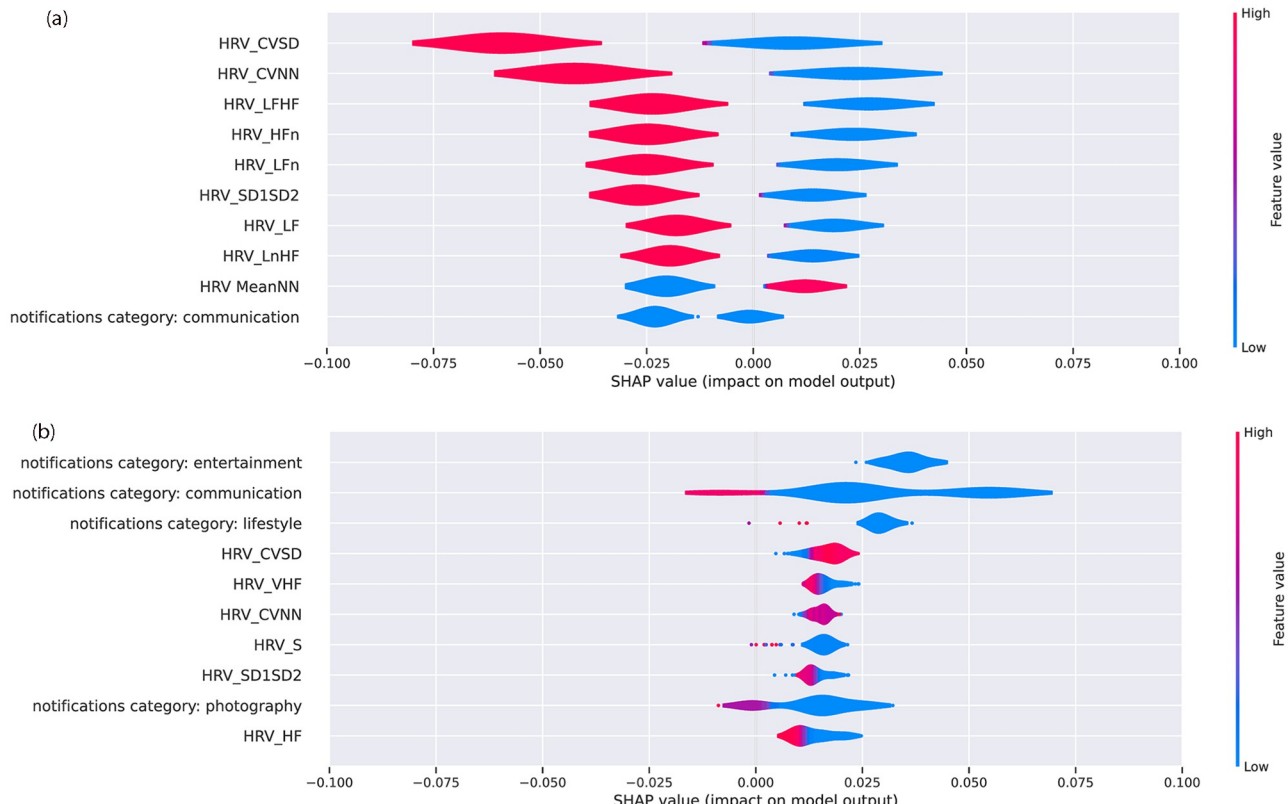

**Fig 5. Summary of SHAP-based explainers for top 10 features for Participants 8 (a) 10 (b).** HRV data are extracted from watch signals. HRV_CVSD is RMSSD divided by the mean of the RR intervals (MeanNN). HRV_LF and HRV_HF are the spectral power of low frequencies (0.04 to 0.15 Hz) and high frequencies (0.15 to 0.4 Hz) respectively. HRV_LFn and HRV_HFn are power-normalized versions of HRV_LF and HRV_HF features. SDNN the standard deviation of the RR intervals. CVNN is the standard deviation of the RR intervals (SDNN) divided by the mean of the RR intervals. Please refer to S1–S3 Tables in S1 Appendix for the full list of features description.

which were trained with data collected from a single device (i.e., smart ring, smart watch, or smartphone), and one of which was trained using data from all three devices. Subsequently, the models were evaluated using test data from the same device(s). This assessment strategy allowed us to select the optimal set of devices based on the desired criteria, including performance, costs, or user burden. The performance of the models is indicated in Table 2. The smart ring (Oura) is small, lightweight, and easy-to-use, with a battery life of approximately one week following a full charge. The device provides sleep quality, physical activity, and nocturnal HR and RMSSD parameters with a rather high accuracy [40, 60, 61]. Although Oura obtained the highest recall value compared to all other cases, it showed the poorest accuracy, F1-score, and precision for loneliness detection. The second-best accuracy of a single device was obtained using the smart watch (Samsung), which enabled us to acquire HR and multiple HRV measurements at all times. In comparison to Oura, the watch obtained better precision but worse recall. The AWARE framework, which only uses smartphone logging and sensing features, obtained the highest accuracy and precision but the lowest recall compared to the Oura ring and Samsung watch. AWARE runs as a background application on the user's smartphone. This passive sensing eliminates the need for user input, in contrast to wearable devices that require continuous wear. It should be noted however, that the AWARE data collection is limited to the behavioral and contextual parameters. For instance, our previous study [62]

indicated that COVID-19 lockdown had an adverse impact on such passive smartphone-based data collection, as user mobility decreased (e.g., fewer location changes). Our results showed that multimodality has a positive impact on the overall performance of loneliness detection, providing an improved trade-off between precision and recall. Specifically, the three devices used in the study together obtained higher precision values compared to the Oura ring alone, and slightly higher recall values compared to Samsung Watch and AWARE alone.

## Discussion

Advances in technology offer exciting future direction for mental health prevention and treatment. With the availability of passive sensing and assessments of people's experiences, behaviors, and mental states in real-time and with greater ecological validity, researchers and medical professionals can have a better understanding of individualized experiences of mental health [63]. Loneliness is an important predictor of physical health and mortality [64]. Furthermore, loneliness is associated with several sleep and physiological factors [65]. The U.S. has previously announced a loneliness epidemic [66] and several recent studies are focusing on loneliness across different populations [67].

Various external influences, both on a societal and individual level, can affect one's emotional well-being. Historical factors, such as the cyclical stress from school during exam seasons and the challenges students faced when readjusting to face-to-face instruction, come into play. Those who were reminded of their personal connections might have felt less isolated during such transitions. It is also important to consider the time period that the study took place along with each study period and how large events may have impacted the participants (i.e. changes in UCI on/off-campus schedules due to COVID-19, midterms and finals, mass shootings, or holidays such as Chinese New Year). These events and others could lead participants to feel less connected with others over time, especially with friends and family.

To address loneliness requires a multimodal approach. Our study adopted a comprehensive approach using multiple devices to passively and continuously monitor people's experiences in order to predict loneliness and assess how loneliness might look across individuals.

Our findings in this study showed that loneliness detection using multimodal assessment with commercially available devices could yield an accuracy of 82%. We compare our proposed ML-based loneliness detection method with current state-of-the-art methods for fully and objectively (i.e., only using sensors from wearable and mobile devices) identifying loneliness. To the best of our knowledge, there exists only one article in the literature which attempts to objectively detect instantaneous loneliness (i.e., not daily or weekly classification) while having an acceptable sample size ($N \geq 10$). It should be noted that in order to utilize a predictive model in just-in-time adaptive interventions (JITAI) [68], it is critical to deploy models with the ability of instantaneous (fine-grained) predictions. Wu et. al. [37] conducted a three-week data collection of 129 individuals using smartphones. The EMA was collected randomly up to four times a day in which participants were asked to select 4 options of loneliness levels. Using a random forest classifier, they obtained the average AUC of 0.73–0.74 for loneliness detection, which is 10% lower than our method's AUC. Although their results showed how contextual data collected from smartphone sensors could be used in loneliness detection, they did not leverage physiological measurement and multi-modal assessment in their study. As presented in the following, physiological indicators are rich sources of information for loneliness detection models when coupled with contextual markers.

With respect to feature importance for predicting loneliness, we found that across devices, assessments of people's phone behaviors using the AWARE app had the highest average values with loneliness across all participants. More specifically, people's use of their phones either

through notifications or engagement with phone apps and calls were highly associated with loneliness. In this respect, our findings do support the existing literature that has primarily used mobile app assessments of loneliness using people's geolocation and phone interactions with others [32, 37, 38]. HRV parameters collected through the Samsung smartwatch were second in their prediction weights on loneliness.

Surprisingly, sleep measures as assessed through the ring had less weight as predictors of loneliness. This finding is somewhat inconsistent with existing literature. Past studies that have used subjective indices of sleep suggest a link between reported sleep disturbances and loneliness [45, 46]. Furthermore, studies that have used objective indicators of sleep (such as polysomnography, sleep watches, and neural scans) are consistent with subjective indices, where sleep quality and deprivation was also found to be associated with social withdrawal and loneliness [69]. One potential limitation of our work that may help to explain our finding is that our model included several features from assessments collected during the day, whereas the Oura ring was the only device that captured bedtime features. While behavioral and some of the activity data are being generated throughout the day, sleep data has values per day. This difference in the frequency of sleep data could suggest further exploration on specific data fusion techniques.

With respect to the nuances across behavioral features and cardiovascular features, one possible explanation for why behavioral features weighed more heavily is the growing literature examining the importance of context [70–72]. For example, the links across emotions, physiological assessments, and health vary widely across different racial/ethnic groups [73, 74]. Assessing these objective markers lack meaning if not measured with the context of the individual. Loneliness is highly related to a person's perceived social interactions or lack thereof. If an individual feels slight changes in their interactions with others, whether it be in their use of phone apps or virtual engagement with others, it might have changes in their loneliness. Therefore, in having a more comprehensive approach to understanding how loneliness occurs, it still is needed to ground it in the context of social interactions and relationships. While our sample represented a relatively diverse group of college students, future study may consider not only recruiting a larger sample but also exploring these nuances in behavioral features across other diverse groups. We focused on emerging adulthood given the pervasiveness of smartphone and technology use and the recent literature on the rise of loneliness in this subpopulation [75]. However, loneliness is also a concern in older adults, and so it would be beneficial to examine the objective predictors of loneliness in older adults as well.

Our aim was to obtain the best performance for objective loneliness detection using all available devices. In other words, in addition to assessing the performance of loneliness detection methods, it is vital to evaluate how each device performs individually. Through this process, researchers can gain insights on how to improve other metrics, such as feasibility and usability within the context of remote health monitoring studies. This information can assist researchers in selecting the optimal set of devices according to their specific requirements. Our findings indicate that multimodality yields a beneficial effect on the overall performance of loneliness detection, offering an enhanced balance between precision and recall. Additionally, multimodality offers competitive data fusion [76] in loneliness detection, which can enhance service availability. This enhancement is achieved through the utilization of three independent battery-powered devices, which ensures that loneliness detection remains operational even if one or two of the devices become unavailable due to factors such as battery depletion, technical problems, or decreased user mobility. Moreover, multimodality provides complementary fusion of sensing modalities [76], which can improve the explainability of the analysis. Each device is not directly dependent on other devices, but the data can be aggregated to provide a more complete depiction of loneliness under observation. The Oura ring collects physical

activity and nocturnal health parameters, including sleep quality and HR. The Samsung watch provides nighttime and daytime physiological parameters, including HR and HRV. AWARE enables us to acquire behavioral information (e.g., social interactions) and contextual information (e.g., location). In other words, each device assesses different phenomena, and assessing all of these at the same time, in real-time, can provide a more comprehensive view of how loneliness occurs for each individual. By including each device in analyses, a potentially more accurate examination of associations between loneliness and diverse health and well-being parameters can be tested. Therefore, it can provide holistic actionable insights for health providers. Overall, multimodality is a promising approach for improving the performance, availability, and explainability of loneliness detection methods, however, it increases the cost and user burden.

## Limitations and future directions

Despite this growing concern surrounding loneliness, similar to affect and emotional states—particularly negative ones, loneliness is part of the human experience [3, 4]. It would be naive to simply suggest that we are finding a way to predict loneliness to absolve it in its entirety. However, in finding ways to objectively predict and monitor loneliness, we may be able to help individuals and their clinical providers become aware of the contexts and behaviors that particularly elicit feelings of loneliness. Doing so aligns with prior theory that awareness offers a first step to acceptance or enacting any behavioral change in a way that fits with the individual's goal [77, 78]. It can help individuals view their situations as less lonely, or to reduce the negative feelings that arise in states of loneliness. Our work altogether builds upon and contributes to the field of precision medicine and individualized approaches to acknowledge that an individual's own contexts and way they perceive their experiences matter in addressing mental health [63, 79].

Loneliness is associated with several other negative feelings and emotional states, including anxiety and depression [2, 4, 80]. While the focus of our study was considering the ways we could objectively measure behavior and experiences tied to loneliness, it could also be beneficial to consider other subjective states that are experienced alongside feelings of loneliness. The associations between loneliness and other emotions may play a role in the kinds of behaviors and physiological experiences individuals have that we aimed to have collected.

The method we propose also faces technical challenges and can pave the way for subsequent research endeavors. Mobile devices like smartphones and smartwatches have battery constraints, and continuous mental health monitoring can deplete them rapidly. Regular data synchronization may also use significant network bandwidth, which can become costly, especially during roaming. These issues indicate the need for refining the proposed approach for greater feasibility. Potential refinements might include optimizing data collection, processing, and inference in a decentralized manner.

## Conclusion

In this paper we proposed a method for detecting loneliness in a multimodal fully objective setup using smart phones and commercially available wearable devices. We tested our method in a 2-month long study and showed that we can detect loneliness in a continuous way. Our results showed that smartphones and activity-related information during the day are among the most important features for most of the participants. Leveraging multi-modal assessments and passive sensing to continuously and objectively monitor human experiences can help reduce participant burden and support on-going efforts to design personalized treatment and programming to support well-being. Assessing human experiences in real-time can allow us to

take a more preventative approach for health treatment rather than a reactive approach. The work conducted in this study was restricted to a demographic of college students and limited to a 2 month period, which therefore suggests further investigation for confirming the generalizability. We are planning to further investigate device-specific feature extraction techniques and personalized modeling for our proposed method.

## Supporting information

**S1 Appendix. Present the comprehensive list of extracted features from each modality along with descriptions elucidating their respective representations.**
(PDF)

## Author Contributions

**Conceptualization:** Salar Jafarlou, Sina Labbaf, Jessica L. Borelli, Nikil D. Dutt, Amir M. Rahmani.

**Data curation:** Salar Jafarlou, Yuning Wang, Sina Labbaf, Nikil D. Dutt, Amir M. Rahmani.

**Formal analysis:** Salar Jafarlou, Yuning Wang, Jessica L. Borelli, Nikil D. Dutt, Amir M. Rahmani.

**Funding acquisition:** Jessica L. Borelli, Nikil D. Dutt, Amir M. Rahmani.

**Investigation:** Salar Jafarlou, Yuning Wang, Jessica L. Borelli, Nikil D. Dutt, Amir M. Rahmani.

**Methodology:** Salar Jafarlou, Jocelyn Lai, Yuning Wang, Jessica L. Borelli, Nikil D. Dutt, Amir M. Rahmani.

**Project administration:** Salar Jafarlou, Jocelyn Lai, Jessica L. Borelli, Nikil D. Dutt, Amir M. Rahmani.

**Resources:** Salar Jafarlou, Yuning Wang, Brenda Nguyen, Hana Qureshi, Christopher Marcotullio, Jessica L. Borelli, Nikil D. Dutt, Amir M. Rahmani.

**Software:** Salar Jafarlou, Yuning Wang, Sina Labbaf, Nikil D. Dutt, Amir M. Rahmani.

**Supervision:** Jessica L. Borelli, Nikil D. Dutt, Amir M. Rahmani.

**Validation:** Salar Jafarlou, Jessica L. Borelli, Nikil D. Dutt, Amir M. Rahmani.

**Visualization:** Salar Jafarlou, Yuning Wang, Amir M. Rahmani.

**Writing – original draft:** Salar Jafarlou, Iman Azimi, Jocelyn Lai, Yuning Wang, Brenda Nguyen, Jessica L. Borelli, Nikil D. Dutt, Amir M. Rahmani.

**Writing – review & editing:** Salar Jafarlou, Iman Azimi, Jocelyn Lai, Brenda Nguyen, Hana Qureshi, Christopher Marcotullio, Jessica L. Borelli, Nikil D. Dutt, Amir M. Rahmani.

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
