## [Decision Letter · Decision Letter 0]

24 Aug 2023

PONE-D-23-21317Objective Monitoring of Loneliness Levels using Smart Devices: A Multi-Device Approach for Mental Health ApplicationsPLOS ONE

Dear Dr. JAFARLOU,

Thank you for submitting your manuscript to PLOS ONE. After careful consideration, we feel that it has merit but does not fully meet PLOS ONE’s publication criteria as it currently stands. Therefore, we invite you to submit a revised version of the manuscript that addresses the points raised during the review process.

We look forward to receiving your revised manuscript.

Kind regards,

Liang-Chih Chang

Academic Editor

PLOS ONE

3. We notice that your supplementary tables are included in the manuscript file. Please remove them and upload them with the file type 'Supporting Information'. Please ensure that each Supporting Information file has a legend listed in the manuscript after the references list.

Additional Editor Comments:

The manuscript is well-written with a good logical flow which allows readers to understand the complicated research design easily. I’d appreciate more information in some areas that could help other researchers replicate the study and support the utilization of the research findings in practice.

1. The authors have suggested that there were relatively few studies using the objective ways to collect information regarding loneliness. It’s important to discuss why it is important to examine which devices can provide accurate information in predicting levels of loneliness in terms of its research significance as well as clinical significance.

2. The authors indicated the period for their participants recruitment. Can they also specify the period of time for their data collection? Did they use “rolling enrollment” (i.e., each participant was engaged in the experiment immediately when they signed up for the study)? Or did every participant start and end the experiment at the same time?

3. Can the authors offer more information on how they collect subjective experiential data? For example, besides asking participants to rate their levels of loneliness, did they collect other information?

4. Can the authors report the percentage of missing data? I am curious to know, for example: If the participants missed a significant portion of data collection points, such as 50%, would it be valid to replace the missing data using the average?

5. When collecting the behavioral data (e.g., the usage of different apps), did the authors also collect their usage on other electronic devices (e.g., computer, tablet) or ask their preference on devices (i.e., Is the smart phone their main device for social-related apps?)?

6. I found the information on the binary classification labels interesting. Can the authors explain why they use this method in determining the three categories: accuracy, precision, and recall? Also, what do these three categories actually mean in terms of data interpretation?

7. Can the authors explain how their research findings can be utilized in practice? In other words, can practitioners help their clients detect loneliness using smart devices? Also, what is the clinical significance of knowing the objective ways of measuring loneliness?

8. The study was conducted during the pandemic. Can the authors discuss the potential impact of the pandemic on their research findings?

Reviewers' comments:

Reviewer's Responses to Questions

**Comments to the Author**

1. Is the manuscript technically sound, and do the data support the conclusions?

Reviewer #1: Yes

2. Has the statistical analysis been performed appropriately and rigorously? 

Reviewer #1: Yes

3. Have the authors made all data underlying the findings in their manuscript fully available?

Reviewer #1: No

4. Is the manuscript presented in an intelligible fashion and written in standard English?

Reviewer #1: Yes

5. Review Comments to the Author

Reviewer #1: This study provides valuable insights into digital technology, utilizing wearable devices and smartphones to collect physiological signals, sleep patterns, and behavioral information such as social interactions. It continuously and reliably collected extensive behavioral data from 30 participants over two months through a multimodal approach. SHapley Additive exPlanations (SHAP) analysis further enhances the understanding of the collected data. This research offers a more comprehensive monitoring technique for future lifestyles. The article is engaging and merits consideration for publication; however, minor revisions should indicate some concerns. Suggestions are as follows:

1. In the paper, the reference to the loneliness experienced by individuals during the COVID-19 pandemic, along with collecting participants' daily behaviors through technological devices, highlights the behaviors that might influence loneliness. However, before delving into this topic, it is essential to define and differentiate between 'aloneness' and 'loneliness,' and provide relevant citations. Moustakes (1961) approached this concept from a philosophical perspective, considering loneliness an inherently human experience. On the other hand, Peplau & Perlman (1982) compiled sociologists' definitions of loneliness as an emotional response to social deficiency. They emphasized that individuals should learn to coexist with loneliness due to experiences related to social deficiency.

2. To clearly define 'loneliness' in this study, it is necessary to distinguish it from similar emotions. This will facilitate a deeper understanding and precise definition of loneliness in the context, emphasizing that loneliness is not synonymous with 'aloneness' or 'isolation.' Furthermore, establishing dimensions for measuring loneliness requires a comprehensive approach, and it is essential to refer to relevant literature to explore the relationship between loneliness and other emotions. For instance, McWhirter (1990) observed from clinical reports that feelings of loneliness are correlated with other emotions such as depression, anxiety, and passiveness. However, the causal relationship between these emotions remains unclear. This study's contribution can be emphasized by highlighting how it aligns with previous research on scholars' attempts to mitigate loneliness through consumer behavior patterns.

3. By emphasizing the contributions of this study, it is worth noting previous scholars' findings. Sullivan (1982) proposed that individuals may not be aware of their own or others' states of loneliness or social deficiency. Young (1982) highlighted the temporal aspect of loneliness, suggesting that studies that dynamically collect individual information over extended periods offer exceptional reference value. Rook (1984) defined loneliness as a continuous state of emotional distress that occurs in situations of isolation, misunderstanding, or rejection, where there is a lack of appropriate social interaction and engagement in hopeful activities. Relief can be obtained through opportunities for emotional closeness. Therefore, this study can effectively assess and suggest the improvement of loneliness through a 'Loneliness Index,' encouraging participants to engage in appropriate social interactions to alleviate loneliness effectively.

4. This study should address the potential impact of different racial and ethnic groups on the relationships between emotions and physiological indicators, as highlighted by Mesquita et al. (2017) and Williams (2009) [67,68]. To mitigate these potential effects, it is important to clarify the demographic characteristics of the 30 participants, including their ethnic backgrounds and genders. Furthermore, it is crucial to note that loneliness is not exclusive to college students, various age groups, especially the elders are likely to happen on them. Therefore, it is essential to elaborate on why participants between 18 and 22 were selected and explain the rationale behind excluding older participants in this study.

5. This study addressed an assessment strategy to select the optimal set of devices based on the desired criteria. In the results, the loneliness detection models obtained an accuracy of 82% in Table 2, precision score of 81%, and value of 0.83 Area Under Curve (AUC) as well. However, it is lack of comparing the advantage and disadvantages of the proposed methods. It needs to describe more detailed about your proposed methods.

6. PLOS authors have the option to publish the peer review history of their article (what does this mean?). If published, this will include your full peer review and any attached files.

Reviewer #1: No

---

## [Author Response · Author response to Decision Letter 0]

29 Jan 2024

We have put the response to academic editors and reviewers in a separate pdf document as instructed.

---

## [Editor Report · Decision Letter 1]

2 Feb 2024

Objective Monitoring of Loneliness Levels using Smart Devices: A Multi-Device Approach for Mental Health Applications

PONE-D-23-21317R1

Dear Dr. JAFARLOU,

We’re pleased to inform you that your manuscript has been judged scientifically suitable for publication and will be formally accepted for publication once it meets all outstanding technical requirements.

Kind regards,

Liang-Chih Chang

Academic Editor

PLOS ONE
---

## [Editor Report · Acceptance letter]

10 May 2024

PONE-D-23-21317R1 

PLOS ONE

Dear Dr. Jafarlou, 

I'm pleased to inform you that your manuscript has been deemed suitable for publication in PLOS ONE. Congratulations! Your manuscript is now being handed over to our production team.

Kind regards, 

on behalf of

Dr. Liang-Chih Chang 

Academic Editor

PLOS ONE